# Mutant IDH1 Differently Affects Redox State and Metabolism in Glial Cells of Normal and Tumor Origin

**DOI:** 10.3390/cancers11122028

**Published:** 2019-12-16

**Authors:** Julia Biedermann, Matthias Preussler, Marina Conde, Mirko Peitzsch, Susan Richter, Ralf Wiedemuth, Khalil Abou-El-Ardat, Alexander Krüger, Matthias Meinhardt, Gabriele Schackert, William P. Leenders, Christel Herold-Mende, Simone P. Niclou, Rolf Bjerkvig, Graeme Eisenhofer, Achim Temme, Michael Seifert, Leoni A. Kunz-Schughart, Evelin Schröck, Barbara Klink

**Affiliations:** 1Institute for Clinical Genetics, Faculty of Medicine Carl Gustav Carus, Technische Universität Dresden, Fetscherstraße 74, 01307 Dresden, Germany; julia.biedermann@ukdd.de (J.B.); matthias.preussler@ukdd.de (M.P.); khalilard@gmail.com (K.A.-E.-A.); Alexander.Krueger@ukdd.de (A.K.); Evelin.Schrock@tu-dresden.de (E.S.); 2Department of Neurosurgery, University Hospital Carl Gustav Carus, Technische Universität Dresden, 01307 Dresden, Germany; marinacondeperez@outlook.com (M.C.); Ralf.Wiedemuth@ukdd.de (R.W.); Gabriele.Schackert@ukdd.de (G.S.); Achim.Temme@ukdd.de (A.T.); 3Institute of Clinical Chemistry and Laboratory Medicine, University Hospital Carl Gustav Carus, Technische Universität Dresden, 01307 Dresden, Germany; Mirko.Peitzsch@ukdd.de (M.P.); Susan.Richter@ukdd.de (S.R.); Graeme.Eisenhofer@ukdd.de (G.E.); 4OncoRay-National Center for Radiation Research in Oncology, Faculty of Medicine and University Hospital Carl Gustav Carus, Technische Universität Dresden and Helmholtz-Zentrum Dresden-Rossendorf, 01307 Dresden, Germany; Leoni.Kunz-Schughart@oncoray.de; 5National Center for Tumor Diseases (NCT), Partner site Dresden, 01307 Dresden, Germany; michael.seifert@tu-dresden.de; 6German Cancer Consortium (DKTK), Dresden, German Cancer Research Center (DKFZ), 69120 Heidelberg, Germany; 7Institute for Pathology, University Hospital Carl Gustav Carus, Technische Universität Dresden, 01307 Dresden, Germany; Matthias.Meinhardt@ukdd.de; 8Department of Biochemistry, Radboud University Medical Center, 6525 Nijmegen, The Netherlands; William.Leenders@radboudumc.nl; 9Experimental Neurosurgery, Department of Neurosurgery, University Hospital Heidelberg, 69120 Heidelberg, Germany; Christel.Herold-Mende@med.uni-heidelberg.de; 10Department of Oncology, NorLux Neuro-Oncology Laboratory, Luxembourg Institute of Health (LIH), L-1526 Luxembourg, Luxembourg; simone.niclou@lih.lu (S.P.N.); Rolf.Bjerkvig@uib.no (R.B.); 11Department of Biomedicine, University of Bergen, 5020 Bergen, Norway; 12Department of Medicine III, University Hospital Carl Gustav Carus, Technische Universität Dresden, 01307 Dresden, Germany; 13Institute for Medical Informatics and Biometry, Faculty of Medicine Carl Gustav Carus, Technische Universität Dresden, 01307 Dresden, Germany; 14National Center of Genetics (NCG), Laboratoire national de santé (LNS), L-3555 Dudelange, Luxembourg

**Keywords:** IDH-mutation, IDH1, glioma, redox state, metabolism, NAD-synthesis

## Abstract

IDH1^R132H^ (isocitrate dehydrogenase 1) mutations play a key role in the development of low-grade gliomas. IDH1^wt^ converts isocitrate to α-ketoglutarate while reducing nicotinamide adenine dinucleotide phosphate (NADP^+^), whereas IDH1^R132H^ uses α-ketoglutarate and NADPH to generate the oncometabolite 2-hydroxyglutarate (2-HG). While the effects of 2-HG have been the subject of intense research, the 2-HG independent effects of IDH1^R132H^ are still ambiguous. The present study demonstrates that IDH1^R132H^ expression but not 2-HG alone leads to significantly decreased tricarboxylic acid (TCA) cycle metabolites, reduced proliferation, and enhanced sensitivity to irradiation in both glioblastoma cells and astrocytes in vitro. Glioblastoma cells, but not astrocytes, showed decreased NADPH and NAD^+^ levels upon IDH1^R132H^ transduction. However, in astrocytes IDH1^R132H^ led to elevated expression of the NAD-synthesizing enzyme nicotinamide phosphoribosyltransferase (NAMPT). These effects were not 2-HG mediated. This suggests that IDH1^R132H^ cells utilize NAD^+^ to restore NADP pools, which only astrocytes could compensate via induction of NAMPT. We found that the expression of NAMPT is lower in patient-derived IDH1-mutant glioma cells and xenografts compared to IDH1-wildtype models. The Cancer Genome Atlas (TCGA) data analysis confirmed lower NAMPT expression in IDH1-mutant versus IDH1-wildtype gliomas. We show that the IDH1 mutation directly affects the energy homeostasis and redox state in a cell-type dependent manner. Targeting the impairments in metabolism and redox state might open up new avenues for treating IDH1-mutant gliomas.

## 1. Introduction

Diffuse gliomas are the most prevalent brain tumors in adults with an overall unfavorable prognosis. Low-grade gliomas and secondary glioblastomas are characterized by mutations in IDH1/2 (isocitrate dehydrogenase 1/2), which are associated with a longer overall survival and better response to therapy compared to IDH-wildtype gliomas [1,2,3,4]. The most frequent mutation observed in more than 90% of cases is the amino acid substitution arginine to histidine at codon 132 of the cytoplasmic enzyme isocitrate dehydrogenase 1 (IDH1 c.395G>A; R132H) [1,2]. Other rare IDH1 variants in codon 132 have been described in gliomas, including R132C, R132S, R132G, and R132L, which show similar effects on IDH1 activity [5]. IDH1/2 mutations are assumed to be one of the earliest genetic alterations and are considered to play a key role in glioma development [6]. However, the mechanisms by which mutant IDH transforms the cell remain insufficiently understood. The wildtype IDH1 converts isocitrate to α-ketoglutarate (α-KG) while reducing nicotinamide adenine dinucleotide phosphate (NADP^+^) to NADPH. In contrast, mutant IDH1 is unable to efficiently catalyze the oxidative decarboxylation of isocitrate, but acquires a neomorphic catalytic activity that allows the NADPH-dependent reduction of α-KG into 2-hydroxyglutarate (2-HG) [7]. 

Now accepted to be an oncometabolite, 2-HG has been a focus for IDH1/2 research for the past decade [8,9]. It has been shown that 2-HG inhibits α-KG-dependent dioxygenases such as methylcytosine dioxygenase (TET2), which leads to the CpG island-methylator phenotype (G-CIMP) prevalent in gliomas with IDH1/2 mutations [10,11]. On the other hand, the supraphysiological accumulation of 2-HG is also seen in patients with 2-hydroxyglutaric acidurias. This rare neurometabolic disease is related to the germline mutations in D2HGDH but, to the best of our knowledge, these patients show no increased prevalence for brain tumor formation [12,13]. This suggests 2-HG independent effects of IDH1^R132H^ during tumorigenesis. 

Recent studies have increasingly focused on the impact of the IDH1^R132H^ mutation on phospholipid synthesis, glutamine metabolism, and the cellular response to treatment [14,15,16,17]. Wildtype IDH1 is reported as the major generator of cytosolic NADPH in glia cells and glioblastoma [18], while IDH1^R132H^ consumes NADPH. The proposed IDH1^R132H^-associated drop in intracellular NADPH levels is expected to have drastic effects on the cellular metabolome, the redox state of the cell, and various other physiological phenomena [19]. 

This study aimed to unravel the 2-HG independent effects of mutant IDH1 on the redox state, metabolism, and energy homeostasis in both tumor and non-neoplastic glial cells. To this end, we transduced glioblastoma cell lines and immortalized astrocytes with the most frequent IDH1^R132H^ mutation for systematic comparative examinations, and also used empty vector control cell lines supplemented with membrane-permeable 2-HG. We show that IDH1^R132H^ has differing effects on the redox state between tumor cells and astrocytes, indicating that IDH1^R132H^ could have different effects depending on the stage of malignant transformation. Our findings might have important implications for future therapies aiming at tumor metabolism in IDH-mutant gliomas.

## 2. Results

### 2.1. Stable Transduction of IDH1^R132H^ in Glioblastoma Cells and Immortalized Astrocytes

We stably transduced U87-MG, the primary patient-derived glioblastoma cell line HT7606, and immortalized astrocytes SVGp12 with *IDH1^R132H^*, *IDH1^wt^*, or an empty vector. For each cell line and vector, we performed three independent transductions that were used throughout all of the experiments with three technical replicates per transduced cell line. The presence of the IDH1^R132H^ mutation on the DNA and RNA-levels in U87-MG-IDH1^R132H^, HT7606-IDH1^R132H^, and SVGp12-IDH1^R132H^ was confirmed using Sanger sequencing (Appendix A). All of the cell lines transduced with IDH1^R132H^ or IDH1^wt^ showed a strong expression of the transduced protein and expression of the endogenous IDH1 as confirmed with specific antibodies using Western Blot and immunocytochemistry, while cells transduced with an empty vector only showed an expression of the endogenous IDH1 (Appendix A). To verify the functionality of the IDH1^R132H^ protein, we measured the intracellular 2-HG levels using LC-MS/MS [20,21]. The cells expressing IDH1^R132H^ had a 1000 to 3000-fold higher intra-cellular concentration than the empty vector controls in all cell models (Appendix A). We could mimic these intracellular 2-HG levels in empty vector control cells by adding 1 mM D-2-HG acid octyl ester sodium salt (referred to as 2-HG from here on), which was used as a further control condition in the following experiments. 

### 2.2. IDH1^R132H^ but not 2-HG Treatment Alone Leads to Changes in TCA Cycle Metabolites 

We applied LC-MS/MS to quantitatively compare the intracellular TCA (tricarboxylic acid) cycle metabolites. The transduction with IDH1^R132H^ caused a general drop in the concentrations of metabolites downstream of isocitrate (α-KG, succinate, fumarate, and malate) in glioma cells and astrocytes, while isocitrate levels increased. In addition, the intracellular level of the amino acid glutamate, which might be used by the cells to restore α-KG levels [22], was reduced in all three cell lines (Figure 1a). On the other hand, the level of the amino acid aspartate increased in the IDH1^R132H^ mutated cells. Aspartate, as glutamate, can be used in anaplerotic reactions to replenish TCA cycle intermediates.

In contrast, cells transduced with IDH1^wt^ had significantly reduced citrate and isocitrate levels, while the α-KG levels were increased compared to the empty vector control cells (Figure 1b). The treatment of the empty vector controls with external 2-HG for 24 h resulted in highly elevated intra-cellular 2-HG levels comparable to IDH1^R132H^-transduced cells but was not accompanied by a significant change in the concentrations of the TCA cycle metabolites (Figure 1c). This indicates that IDH1^R132H^ affects cell metabolism due to either the insufficient conversion of isocitrate to α-KG or the persistent consumption of α-KG for 2-HG production, independent of the 2-HG-level elevation.

### 2.3. IDH1^R132H^ Inhibits Growth and Enhances Radio-Sensitivity In Vitro

Glioma patients with *IDH1* mutations have a longer overall survival and show a better response to treatment; the reasons for this are still unclear. Therefore, we wanted to evaluate the impact of the IDH1^R132H^ on growth and radio-sensitivity in our cell models. The tumor cell lines U87-MG and HT7606 exhibited similar 2-D growth kinetics with doubling times of 33.2 h (±5.5 SD) and 33.2 h (±2.2 SD), respectively. The immortalized astrocytes SVGp12 grew considerably slower, with a doubling time of 60.8 h (±10 SD). In contrast to the U87-MG cell line model, the HT7606-IDH1^R132H^ and SVGp12-IDH1^R132H^ cells showed a significant decrease in viability in vitro compared to both their empty-vector and IDH1^wt^ counterparts (Figure 2a). In line with this observation, the cell numbers were lower in these cultures 72 h after seeding (Figure 2b). However, the colony formation capacity was either unaltered (SVGp12 IDH1^R132H^ vs. empty vector: 2.1% ± 0.1 vs. 3.4% ± 1.9, *p* = 0.31, *t*-Test) or enhanced in HT7606-IDH1^R132H^ compared to the empty vector cells (14.8% ± 4.2 vs. 6.0% ± 1.4; *p* < 0.05, *t*-Test) (Figure 2c), implying that the observed reduction in the cell number and viability in IDH1 mutated cells might indeed be due to reduced proliferation. Treatment with 2-HG alone did not lead to a similar decrease in proliferation in glioblastoma cell lines, but slightly reduced proliferation of SVGp12 astrocytes. 

There was no difference in proliferation of U87-MG-IDH1^R132H^ compared to IDH1^wt^ cells in 2-D culture growth; however, the colony forming capacity was also increased in the U87-MG-IDH1^R132H^ compared to the empty-vector control (10.8% ± 0.6 vs. 8.0% ± 0.1; *p* < 0.05 using *t*-Test) (Figure 2c). U87-MG cells are known to also grow in 3-D culture [23]. We therefore included a 3-D assay to verify the biological relevance of the IDH-mutation in this cell line, which appeared unaffected in the 2-D growth setting. After a 4-day initiation interval, spheroids of U87-MG-IDH^R132H^ and the U87-MG control cells reached similar sizes when seeding identical cell numbers/well. Hence, the IDH-mutation did not per se alter spheroid formation (Figure 2d). However, monitoring the spheroid volume as function of time revealed critically reduced volume growth kinetics of the U87-MG-IDH1^R132H^ spheroids (Figure 2d), indicating that the IDH1^R132H^ indeed inhibits growth even in the U87-MG cell line model, but this only manifests in a 3-D environment. 

Clonogenic survival assays showed that the tumor cell lines were intrinsically more radio-resistant than the immortalized astrocytes, with U87-MG exhibiting the lowest radio-sensitivity (Figure 2e). Transduction with mutant IDH1 led to a reproducible and significant radio-sensitization in the glioblastoma cell line U87-MG and astrocytes SVGp12 (*p* < 0.001) (Figure 2e), while the survival curves of vector control and IDH-mutated patient derived cell line HT7606 did not systematically differ. Nonetheless, a clearly reduced clonogenic survival was also observed in the latter upon IDH1^R132H^ transduction for the high radiation doses of 10 Gy. 

### 2.4. Intracellular NADPH Levels Significantly Drop in Glioma Cells but not in Astrocytes Upon Transduction with IDH1^R132H^

In addition to the abolishment of the enzyme’s wildtype function of generating α-KG and providing NADPH, IDH1^R132H^ consumes NADPH to generate 2-HG. In our cell line panel, the basal levels of NADPH and total NADP (NADPt = NADP^+^ + NADPH) were highest in HT7606; U87-MG and SVGp12 exhibited similar amounts of NADPH (Appendix A). We found significantly lower NADPH levels in U87-MG-IDH1^R132H^ and HT7606-IDH1^R132H^ compared to the empty vector control cells (Figure 3a). In contrast, the astrocytes displayed increased intracellular NADPH levels upon IDH1^R132H^ transduction. When looking at the NADPH/NADPt ratio, however, all the cell models, including the astrocytes, showed a shift towards NADP^+^ (Figure 3a). Membrane permeable 2-HG did not alter the NADPH/NADPt ratios, indicating that the observed shift directly resulted from the neomorphic NADPH-consuming enzymatic activity of IDH1^R132H^. The unexpected increase in NADPH levels in the astrocytes expressing IDH1^R132H^ relates to an overall higher intracellular NADPt pool in these cells (Figure 3a). In contrast, U87-MG and HT7606 showed a decrease in NADPt concentrations upon transduction with IDH1^R132H^. These findings indicate that IDH1^R132H^ can have different effects on NADPt pools in non-neoplastic and neoplastic cells.

### 2.5. IDH1^R132H^ Leads to a Decrease in NAD^+^ and the Activity of NAD-Dependent Enzymes in Glioma Cells but not in Astrocytes

One way to restore essential NADP^+^ and NADPH concentrations in mammalian cells is through the cytosolic enzyme NAD kinase (NADK). NADK utilizes ATP to phosphorylate NAD^+^ to NADP^+^ [24,25]. Indeed, a significant drop in NAD^+^ and NADt concentrations in HT7606-IDH^1R132H^ and U87-MG-IDH1^R132H^ was seen compared to the respective empty vector control cells (Figure 3b). In contrast, astrocytes with and without IDH1^R132H^ retain relatively stable levels of NAD^+^ and NADt. The enzymatic activity of sirtuins, multifunctional NAD^+^-dependent proteins, was measured to validate these results. In line with the previous observations, we detected a significant decrease in the enzymatic activity of sirtuins in the two IDH1^R132H^ glioblastoma cell lines, but not in the astrocytes (Figure 3b). Again, treatment with 2-HG alone had no effect on NAD^+^ and NADt levels or the enzymatic activity of sirtuins in all our cell models. From these data we conclude that IDH1-mutant cells indeed utilize NAD^+^ to restore NADPH levels, but only astrocytes might be able to compensate for the increased NAD^+^ consumption. 

### 2.6. Expression of NAD^+^ Synthesis Enzymes Varies between Individual Cell Lines

In mammalian cells, NAD^+^ can be synthesized from nicotinamide (NAM), nicotinic acid (NA), nicotinamide riboside (NR; salvage pathways) and the essential amino acid tryptophan (de novo synthesis) via four overlapping pathways [26] (Figure 4a). We therefore analyzed the expression levels of the rate-limiting enzymes in these four pathways, namely nicotinamide phosphoribosyltransferase (NAMPT), nicotinic acid phosphoribosyltransferase (NAPRT), nicotinamide riboside kinase 1 (NMRK1), and the de novo synthesis enzyme quinolinic acid phosphoribosyltransferase (QPRT). Additionally, we analyzed 3-Hydroxyanthranilat-3,4-Dioxygenase (3-HAO), the enzyme generating quinolinic acid, the substrate for QPRT. 

The expression of the NAD^+^ synthesis enzymes differed between the three cell models used in our study. While the U87-MG cells showed strong expression of QPRT, HT7606 and SVGp12 had very low enzyme levels on the transcriptional and translational levels (Figure 4b and Appendix A). Besides, the three cell lines neither expressed 3-HAO nor the NAPRT protein, irrespective of their IDH1 mutation status or exposure to 2-HG (Figure 4b and Appendix A). NMRK1 and NAMPT were expressed in all three cell lines. However, the NAMPT protein and mRNA levels were higher in the glioblastoma cell lines than in the astrocytes (Figure 4b and Appendix A).

### 2.7. Different Effect of IDH1^R132H^ on NAMPT-Expression between Glioma Cells and Astrocytes

While the NAMPT protein levels were strongly reduced in U87-MG-IDH1^R132H^ compared to the empty vector controls, transduction with IDH1^R132H^ led to increased NAMPT protein levels in the astrocytes SVGp12 (Figure 4b,d and Appendix A). However, the NAMPT levels in SVGp12-IDH1^R132H^ were still below the NAMPT levels seen in the glioblastoma cells with and without IDH1 mutation (Appendix A). The downregulation of NAMPT in the IDH1^R132H^-overexpressing glioblastoma cell line U87-MG was also evident on the mRNA level (Appendix A). In HT7606, the NAMPT levels did not change upon transduction with IDH1^R132H^. The NMRK1 protein expression was not influenced by IDH1^R132H^ in any of the cell models (Figure 4b and Appendix A).

### 2.8. Expression of NAD^+^ Synthesis Enzymes Varies in Patient-Derived IDH1^R132H^ and IDH1^wt^ Glioma Cells

We further investigated the expression of NAD-synthesis enzymes in a very rare cohort of three patient-derived IDH1-mutant glioma cell models that have native homozygous IDH1^R132H^ and additionally five patient-derived glioblastoma cell lines with IDH1-wildtype status. In accordance with our findings in U87-MG and HT7606, the QPRT expression was variable in these patient-derived glioma cells: 1/3 IDH-mutant and 5/5 IDH-wildtype glioma cell models expressed QPRT (3/5 with high and 2/5 with low QPRT levels) (Figure 4c and Appendix A). Again, none of the cell models expressed NAPRT. As with QPRT, the NMRK1 expression was highly variable between individual glioma cells, with cells displaying no detectable protein levels up to a strong expression. Importantly, NAMPT was ubiquitously expressed in all the patient-derived glioma cell models. Moreover, the IDH1-mutant cells had overall lower protein levels of NAMPT compared to IDH1-wildtype glioblastoma cells (Figure 4c and Appendix A). This difference was more pronounced at the protein level than at the RNA level, indicating that the posttranslational regulation of NAMPT expression differs between IDH1-mutant and IDH1-wildtype cells (Figure 4c,e and Appendix A).

### 2.9. IDH1^R132H^ Gliomas Show Lower Expression of NAMPT In Vivo

To validate our findings in vivo we analyzed the tissue of two previously generated intracranial patient-derived xenografts (PDX) of glioma, one IDH1-wildtype glioblastoma and one oligodendroglioma with the IDH1^R132H^ [17]. While none of the cell lines expressed NAPRT, there was a strong expression of NAPRT in the tumor tissues from PDXs, independent of IDH1-mutation status (Figure 4c). This indicates that NAPRT might not be expressed by the tumor cells but by the normal surrounding tissue (e.g., endothelial cells). NMRK1 was equally expressed in both the IDH1-wildtype as well as in the IDH1^R132H^ PDX tissues, and the QPRT expression was low in these PDX models. Similarly to what we observed in vitro, the PDX with the IDH1^R132H^ mutation had lower protein levels of NAMPT compared to the IDH1-wildtype glioblastoma (Figure 4c and Appendix A). We next validated our findings in publicly available The Cancer Genome Atlas (TCGA) datasets from primary glioblastoma and low-grade astrocytoma tissues and confirmed that *NAMPT*, *QPRT*, and *NMRK1* were significantly higher expressed in IDH1-wildtype primary glioblastomas compared to IDH1-mutant astrocytomas WHO grade II and III (Wilcoxon rank sum tests: *p*-values ranging from 5.6 × 10^−12^ for *NMRK1* to 1.2 × 10^−32^ for *NAMPT*), while *NAPRT* was equally expressed (Figure 4g and Appendix A).

## 3. Discussion

Here, we demonstrate that IDH1^R132H^ has 2-HG independent effects on metabolism, redox state and energy homeostasis, as well as on proliferation. In addition, we show for the first time a different impact of IDH1^R132H^ on NADPH and NAD^+^ levels in glioma cells and astrocytes, suggesting a cell-type- or cancer-stage-dependent effect of the mutation. Importantly, the IDH1^R132H^ mutation influenced the expression of the NAD-synthesis-associated enzyme NAMPT differently in the glioma cells and astrocytes, supporting the hypothesis that the IDH1 mutation differentially affects cells during tumorigenesis. 

The concentrations of TCA cycle metabolites downstream of isocitrate (α-KG, succinate, fumarate, and malate) dropped in glioma cells and immortalized astrocytes upon transduction with IDH1^R132H^. While there is one in vivo study reporting no change in TCA cycle metabolites in tissues from IDH1-mutant gliomas compared to wildtype tumors [28], our results are in line with other in vitro data [29] and a recently published in vivo study showing that TCA metabolites were reduced in IDH1-mutant patient-derived xenografts and in human glioma tissues [17]. In line with the previous studies [17,28,29], all of our cell models showed significantly reduced glutamate levels upon transduction with IDH1^R132H^. The amino acid glutamate plays an important role as an intermediate metabolite of glutaminolysis. Glutamine is converted to glutamate, which can be subsequently deaminated to α-KG by glutamate dehydrogenase (GDH) and thus replenish the TCA cycle. Therefore, the observed drop in glutamate levels suggests the activation of glutaminolysis in IDH1-mutant cells to compensate and restore α-KG. However, this was not sufficient to maintain normal levels of TCA cycle metabolites in our cell lines. Despite the conversion of glutamate to α-KG via glutamate dehydrogenase (GDH), another possible reaction in the glutaminolytic pathway is the interconversion of glutamate and oxalacetat to aspartate and α-KG via glutamic oxaloactic transaminase (GOT), also called aspartate aminotransferase (AST). Interestingly, in addition to the drop in glutamate levels, we observed enhanced levels of aspartate in all our cell models upon transduction with IDH1^R132H^, which to the best of our knowledge has not been described before. Since aspartate is an additional product of the GOT reaction, one could speculate that enhanced glutaminolysis via GOT might explain the observed increase in aspartate levels and drop in glutamate levels in IDH1-mutant cells. However, further investigations are necessary to understand the mechanisms behind these metabolic changes in IDH1^R132H^ mutated cells.

The addition of 2-HG alone did not affect TCA cycle metabolite or glutamate levels, underlining that the alteration of TCA cycle metabolites is a direct effect of the changed enzymatic function of IDH1^R132H^. The TCA cycle has been implicated as a potential therapeutic target in IDH1-mutant gliomas [30] and our observation that the TCA cycle is compromised in IDH1^R132H^ cells supports this possibility. Altered metabolism in IDH1-mutant tumors can also be used for diagnostic purposes: high levels of 2-HG or decreased levels of glutamate can be detected using in vivo magnetic resonance spectroscopy (MRS) [31,32].

IDH1 mutation but not 2-HG alone led to decreased cell growth in glioblastoma cells as well as astrocytes in vitro. A reduced proliferation rate upon transduction with mutant IDH1 has been reported in glioblastoma cells before [33], but our results suggest that this is independent of 2-HG accumulation. Since all cell lines showed decreased TCA cycle metabolite concentrations upon IDH1^R132H^ transduction, one could speculate that the growth inhibition is mechanistically related to the impaired central carbon metabolism. Interestingly, in spite of the overall growth reduction, the colony forming capacity was enhanced in the IDH1^R132H^ glioblastoma cell lines, indicating that the IDH1 mutation improves clonogenic survival, which might promote tumorigenesis. 

We found an enhanced sensitivity to irradiation in tumor cells and non-neoplastic astrocytes carrying the IDH1^R132H^ mutation. IDH-mutant gliomas are associated with a better prognosis and an increased response to therapy [3,4,34]. Also other in vitro studies showed that over-expressing IDH1^R132H^ led to a better response to radiation in glioma cell lines due to increased oxidative stress [35] while IDH1^R132H^ inhibition increased radio-resistance in cancer cell lines [36]. In addition, silencing IDH1 in glioblastoma cells improved responses to radiotherapy [37]. NADPH is essential for the regeneration of reduced glutathione (GSH), which functions as the main antioxidant in mammalian cells. It has been hypothesized that lower NADPH concentrations lead to decreased GSH levels and increased reactive oxygen species (ROS), and therefore more radiation induced DNA double-strand breaks in IDH1-mutant glioma cells [38]. A recent study in IDH1-mutant patient-derived xenograft models demonstrated that GSH levels were maintained despite altered NADPH/NADP^+^ ratios, but other compensatory pathways for GSH synthesis appeared to be induced [17]. We found that astrocytes also displayed enhanced radio-sensitivity upon IDH1^R132H^-transduction, although they maintained stable NADPH levels, indicating that there might be additional mechanisms involved in IDH1-associated radio-sensitization, which still need to be elucidated. 

IDH1 is reported to be the main generator of NADPH in the brain and in gliomas, producing up to 60% of the cell’s NADPH [18]. The neomorphic enzymatic function of IDH1^R132H^ on the other hand leads to reduction of NADPH to NADP^+^. Accordingly, we observed a shift in the NADPH/NADPt ratio in IDH1-mutant glial cells and tumor cells towards NADP^+^. However, while this was accompanied by a significant drop in absolute NAPDH levels in tumor cells, IDH1^R132H^ transduced astrocytes retained stable NADPH levels due to a compensatory increase in the total NADP pool. 

In mammalian cells, NADP is synthesized from NAD^+^ by NADK [24,25] and the activity of NADK is regulated by NADPH levels, with an increase in NADK activity observed upon a drop in NADPH levels [39]. We therefore hypothesized that IDH1^R132H^-mediated NADPH loss leads to increased NADK activity and NAD^+^ consumption. Indeed, we observed a significant drop in NAD^+^ levels and in the enzymatic activity of NAD^+^-dependent sirtuins upon IDH1^R132H^ transduction in glioblastoma cells, supporting this theory. Treatment with 2-HG alone had no influence, again indicating a direct, non-2-HG-mediated effect of the IDH1 mutation. However, non-neoplastic astrocytes were able to retain stable NAD^+^ levels upon transduction with IDH1^R132H^. 

We found that NAD^+^ restoration in glioblastoma cells and astrocytes relies on the two salvage pathways utilizing nicotinamide riboside and nicotinamide via NAMPT and NMRK1, respectively. In principle, mammalian cells can restore NAD^+^ levels via four different pathways; the key enzymes for each pathway being NAMPT, NAPRT, NMRK1, and QPRT [26]. Of these, only NAMPT and NMRK1 were expressed in all three investigated cell lines. Moreover, NAMPT was the only protein also expressed in each of eight patient-derived IDH-mutant and -wildtype glioma cell models; six of them also showing measureable NMRK1 expression. 

Interestingly, transduction with IDH1^R132H^ led to an increased NAMPT expression in non-neoplastic astrocytes, which had overall low NAMPT levels compared to glioblastoma cells. We therefore conclude that in non-neoplastic cells, NAD^+^-biosynthesis via NAMPT is induced to compensate for the increased NADPH consumption due to IDH1^R132H^ and is sufficient to maintain NAD^+^ homeostasis while increasing the total NADP pool (Figure 5). In contrast, fast proliferating malignant glioblastoma cells can be expected to have a higher demand of NAD^+^, e.g., due to an enhanced turnover or increased activity of NAD-utilizing enzymes. This might explain why glioblastoma cells were not able to compensate for the IDH1^R132H^-mediated metabolic changes, leading to the observed drop in NAD^+^ and NADPH in tumor cells. Indeed, we found, overall, higher levels of NAD^+^-synthesis enzymes NAMPT, NMRK1, and QPRT as well as NADK in primary glioblastomas compared to low-grade gliomas and astrocytes in our and publicly available data, indicating an overall increase in NAD^+^-biosynthesis in malignant cells.

A drop in NAD^+^ concentrations has recently been described in IDH1^R132H^ xenograft-derived cells [40]. The authors hypothesized that this is caused by the downregulation of NAPRT due to hypermethylation. However, our results clearly show that neoplastic and non-neoplastic glial cells completely lack NAPRT expression, independent of IDH1-mutation status. A recent study also demonstrated that, although NAPRT is expressed in the majority of healthy tissues, a significant proportion of glioblastomas are NAPRT deficient [41]. In contrast to cultured cells, we found a strong expression of NAPRT in tissue samples from xenografts, which might be explained by the presence of residual non-glial cells within the tumor tissue. Indeed, publicly available immunohistochemistry data from the Human Protein Atlas (available from www.proteinatlas.org) confirm that NAPRT expression in normal brain and glioma tissue is mainly restricted to endothelial cells [42]. We thus conclude that NAPRT is not involved in NAD^+^ synthesis in gliomas and therefore its downregulation cannot explain the IDH1-related drop in NAD^+^.

Although a previous study described how the malignant transformation in gliomas is associated with a switch in NAD-metabolism towards the de novo synthesis via QPRT [43], we found that QPRT was only expressed in U87-MG, but not in astrocytes or HT7606. The authors also showed that quinolinic acid, the substrate for QPRT, is provided by microglia cells via the enzyme 3-HAO, while this enzyme is not expressed in glioma cells. Our data confirm the complete lack of 3-HAO in non-neoplastic astrocytes and glioma cells, indicating that de novo synthesis from tryptophan does not contribute to NAD-synthesis in our cells.

NAD^+^ pools are involved in the energy homeostasis of the cell, in processes such as DNA repair and telomere maintenance [44], and play an important substrate for ADP-riboyltransferases. NAD^+^-dependent sirtuins, besides their role in deacetylating histones, have a wide variety of roles in the cell, including proper chromosome segregation during the cell cycle, DNA repair, and the regulation of p53 activity [45,46,47]. Therefore, the consumption of NADPH due to IDH1^R132H^ in tumor cells and the drop in NAD^+^ levels, as well as the activity of sirtuins, might affect various processes and should be explored as a potential leverage point for new treatment strategies. Recently, it was shown that NAMPT inhibitors lead to vulnerability in IDH1 mutated brain cancer cells [40]. Although our results indicate that NAMPT is the most important enzyme for NAD-biosynthesis in gliomas, we found its expression to be lower in IDH-mutant gliomas compared to glioblastomas. Moreover, individual tumors can also express NMRK1, or QPRT, or both, which might influence the response to NAMPT inhibitor therapies. Thus, in the future, it might be important to study the influence of the individual expression of NAD^+^ synthesizing enzymes on NAMPT inhibitor therapy. 

It is accepted that the IDH1 mutation occurs very early during gliomagenesis. However, the observation that the IDH1 mutation impairs cell proliferation seems incompatible for a tumor-promoting event. Our results show that IDH1 mutation-mediated effects on redox state and NAD^+^ homeostasis differ between astrocytes and highly proliferating tumor cells. This suggests that the impairment of redox state might only manifest in later tumor stages and, moreover, that slower proliferation might be beneficial for IDH-mutant cells, which provides one possible explanation for the observed slower growth in IDH-mutant gliomas. 

## 4. Materials and Methods

### 4.1. Cells and Cell Culture

All cell lines were cultured in a humidified incubator at 37 °C and 5% CO_2_. The U87-MG (CLS Cell Lines Service, Eppelheim, Germany) cells were maintained in basal medium eagle (BME) with 10% fetal bovine serum (FBS), 10 mM Hepes buffer and 2 mM GlutaMAX™ supplement. The U87-MG identity was verified using spectral karyotyping. The primary glioblastoma cell line HT7606 was obtained in the course of surgery at the Department of Neurosurgery, University Hospital Carl Gustav Carus at the TU Dresden with informed consent and approval of the local ethics committee [48]. The HT7606 cells were cultured in Dulbecco’s modified essential medium (DMEM) with high glucose and GlutaMAX™ supplement, 20% FBS, 10 mM Hepes buffer, and 4× Non-Essential Amino Acids. SVGp12 immortalized astrocytes were purchased from ATCC (ATCC^®^ CRL8621) and cultured in minimum essential medium, alpha modification (αMEM) with GlutaMAX™ supplement and 10% FBS. All media formulations were supplemented with 100 U/mL penicillin and 100 μg/mL streptomycin. All cell culture ingredients except for FBS (Biochrom AG, Berlin, Germany) were obtained from Gibco, Grand Island, NY, USA. In order to investigate the effects of 2-HG, the cells transduced with an empty control vector received 1 mM of (R)-2-hydroxyglutaric acid octyl ester sodium salt (Toronto Research Chemicals, Toronto, Ontario, Canada) in DMEM for 24 h prior to use. 

### 4.2. Patient-Derived In Vitro and In Vivo Models

We analyzed RNA and protein from primary patient-derived IDH1^R132H^ glioma cell lines (NCH5511b, NCH612, NCH1681) and IDH1^wt^ glioma cell lines (NCH644, NCH421k, NCH601, NCH660h, NCH465) [49,50,51]. The cell lines were obtained from patients undergoing surgical resection at the Neurosurgical Department of Heidelberg University Hospital according to the research proposals approved by the Institutional Review Board at the Medical Faculty Heidelberg. Written consent was obtained from each patient. Furthermore, we used RNA and protein from fresh frozen tumor tissue from patient-derived glioma xenografts (PDX) that were generated in NOD/Scid mice from an IDH1^R132H^ glioma (E478 CR500 ‘26’) and an IDH1^wt^ glioma (P3 NC1293 ‘70’) [17,52]. Animal handling and the surgical procedures were performed in accordance with the European Directive on Animal Experimentation (2010/63/EU) and were approved by the institutional and national authorities responsible for animal experiments in Luxembourg (Protocol LRNO-2014-03).

### 4.3. Plasmid Constructs and Lentivirus Production

The coding sequences of *IDH1* (NM_001282386.1) and *IDH1^R132H^* (NM_001282386.1(IDH1):c.395G>A; p.Arg132His), modified by C-terminal His6- and myc-tags, were synthesized (MWG Eurofins Operon, Ebersberg, Germany) and ligated via AgeI and NotI restriction sites into the lentiviral vector pHATtrick-puroR [53], respectively. All vectors were confirmed by DNA sequencing. One established cell line of glioblastoma origin (U87-MG), one primary patient-derived glioblastoma cell line (HT7606) [48], and the immortalized astrocytes (SVGp12) were stably transduced with either IDH1^R132H^, IDH1^wt^, or a pHATtrick-puroR empty vector as previously described [53]. The transduced cells were selected with 15 µg/mL puromycin (Invitrogen, Karlsruhe, Germany) for 24 h.

### 4.4. DNA and RNA Extraction

The DNA from the cultured cells was extracted with the phenol-chloroform extraction method using a Phase Lock Gel (5Prime, Hamburg, Germany). The RNA from the cultured cells was extracted with the miRNeasy mini Kit (Qiagen, Hilden, Germany) according to the manufacturer’s protocol. The DNA and RNA concentrations were measured with a Nanodrop Spectrometer ND1000 (Peqlab, Erlangen, Germany). The RNA quality was evaluated using an Agilent RNA 6000 Nano chip on a 2100 Bioanalyzer. Only RNA with RNA integrity numbers (RIN) of 9–10 were used in further experiments.

### 4.5. PCR and Sequencing

PCR was performed using Qiagen’s HotStarTaq DNA Polymerase kit with primers specific for the exogenous *IDH1* (reverse primer binds on junction of exon 4 and 5) or both exogenous and endogenous copies (reverse primer binds within exon 4) (*IDH1* DNA forward TTGATCCCCATAAGCATGA, *IDH1* DNA and cDNA reverse TCCTGATGAGAAGAGGGTTGA, *IDH1* cDNA forward TTGCTCTGTATTGATCCCCATA). The PCR was performed with the following program: denaturation at 95 °C for 15 min followed by 34 cycles of denaturation at 95 °C for 30 s, annealing at 57 °C for 30 s and extension at 72 °C for 45 s. A final elongation step of 72 °C for 10 min was added. The amplicons were run on a 2% agarose gel impregnated with GelRed at 110 V for one hour and photographed under UV light. The amplicons were sequenced using Sanger’s dye terminator method on the ABI 3130xl DNA Analyzer (Applied Biosystems, Foster City, CA, USA). 

### 4.6. Quantitative Real-Time PCR

The relative expression levels of genes encoding NAD^+^ synthesis enzymes in cell samples were quantified using quantitative real-time PCR. All of the values were normalized to the reference genes *Arf1* and *GAPDH*. The primers are listed in Appendix A. The extracted RNA was reverse transcribed by applying the SuperScriptR VILO™ cDNA Synthesis Kit (Thermo Fisher Scientific, Waltham, MA, USA), and quantitative measurement was performed using the SYBR-Green Mastermix on the 7300 Real Time PCR System (Applied Biosystems) according to manufacturer’s protocols. All of the measurements were done in triplicate. The calculations were made according to the comparative Ct method [54]. 

### 4.7. Protein Extraction and Western Blot

The cultured cells were lysed in RIPA Buffer (Sigma Aldrich, St. Louis, MO, USA) containing a protease inhibitor cocktail (Roche, Basel, Switzerland) according to the manufacturer’s protocol. The extracted proteins were quantified with a Pierce™ BCA Protein Assay Kit (Pierce Biotechnology, Rockford, IL, USA) and resolved on 4–12% Bis-Tris gels (15–30 µg Protein per lane) at 100 V for 30 min. The bands were electrically transferred onto Hybond ECL Membranes (GE Healthcare, Chalfont St Giles, UK) at 30 V for one hour. The membranes were washed once and then blocked in a 5% (w/v) nonfat milk solution. The membranes were probed overnight at 4 °C with shaking with primary antibodies against IDH1^R132H^, IDH1, QPRT, NAPRT1, HAAO, NAMPT and equal loading was ensured by probing with an antibody against GAPDH (for details see Appendix A). Human recombinant proteins NAMPT and HAAO (Abnova, Taipei, Taiwan) were used as positive controls. For visualization, the membranes were probed with horseradish peroxidase-conjugated secondary antibodies, and the Lumi Light PLUS reagent (Roche, Basel, Switzerland) was used for band detection with a chemiluminescence imaging system. The intensity of the Western blot bands was normalized to the GAPDH control bands and quantified using the ImageJ freeware (http://rsb.info.nih.gov/ij/index.html). 

### 4.8. Quantification of TCA Cycle Metabolite Levels Using Liquid Chromatography-Tandem Mass Spectrometry (LC-MS/MS)

For the quantification of the intra-cellular TCA cycle metabolites, 5 × 10^5^ cells/well were seeded into 6-well plates and incubated at 37 °C and 5% CO_2_ for 24 h. The cells were then washed 5 times with PBS on ice and lysed with 100% methanol containing the internal standard mix [20]. After centrifugation, the supernatant was dried and resuspended in 100 µL mobile phase as described in a previous published paper [20]. LC-MS/MS was performed as detailed previously [20]. Briefly, 10 µL samples were injected onto an Acquity UHPLC system (Waters), coupled to an API QTRAP 5500 triple quadrupole mass spectrometer (AB Sciex). A Waters Acquity UPLC^®^ HSS T3 column (1.8 μm, 2.1 × 100 mm) equipped with a guard column facilitated the chromatographic separation using a gradient of the mobile phases A and B which consisted of 0.2% formic acid in water and acetonitrile with a flow rate of 0.459 mL/min. After injection, the initial mobile phase composition of 1% B was kept stable up to 2 min, followed by an increase to 100% B after 2.5 min. After 2.65 min, mobile phase A was increased to reach initial conditions after 3.4 min, followed by a column re-equilibration for another 1.6 min. The multiple reaction monitoring scan mode with negative electrospray ionization was used for quantification.

### 4.9. Colorimetric Measurement of Cellular NAD and NADPH Levels

The intracellular total amounts of NAD^+^ and NADPH were measured using the NAD^+^/NADH and NADP^+^/NADPH Quantification Kits (MBL International Cooperation, Woburn, MA, USA) according to manufacturer’s protocol. In the case of NAD^+^/NADH detection, after the extraction of NAD/NADH with the manufacturer’s extraction buffer, the mixture was filtered through a 10-kDa cut-off filter (Merck Millipore, Billerica, MA, USA) before running the assay. This way, NADH consuming enzymes were removed from the lysate. To detect NADPH only, NADP^+^ was decomposed from the lysate by heating the samples to 60 °C for 30 min. The assays were performed in clear bottom 96-well plates with 50 μL of lysates of 1 × 10^5^ (NADP^+^/NADPH) or 2 × 10^5^ (NAD^+^/NADH) cells per well in triplicates and absorbance was measured at 450 nm using a microplate reader (Tecan, Maennedorf, Switzerland). 

### 4.10. Fluorimetric Measurement of Sirtuin Activity

The activity of intracellular sirtuins was measured using the HDAC Fluorimetric Cellular Activity Assay Kit (Enzo Life Science Inc., Farmingdale, NY, USA) according to the manufacturer’s instructions. Twenty thousand cells per well were seeded in a 96-well plate and grown to confluence overnight. The assay was then performed in phenol-red free IMDM (Gibco, Grand Island, NY, USA) with 10% FBS. The cells were treated with 1 µM Trichostatin to inhibit non-class III HDACs and incubated for 4 h at 37 °C with the Fluor-de-Lys substrate. The cells were then lysed, 2 mM of the sirtuin inhibitor nicotinamide was added, and the plate was read in a fluorimeter (Tecan, Maennedorf, Switzerland) (Ex. 360 nm, Em. 460 nm).

### 4.11. Two-Dimensional Growth Assays

For monitoring the monolayer culture growth, (1–4) × 10^4^ cells were seeded per well into 6-well plates and incubated for up to 9 days. Every 24–48 h cells from three wells were counted upon enzymatic dissociation using a CASY^®^ TTC device (Roche Innovatis AG) and averaged. The growth curves were analyzed by determining the doubling times from the linear regression analysis of the logarithm of the cell counts measured in the exponential phase (N ≥3; n ≥3). The cell numbers of the stably transduced cell lines were compared in short-term cultures by seeding 1 × 10^4^ cells per well into 24-well plates, incubating them for 72 h, and counting them using the CASY^®^ TTC device using medians of three measurements per sample. The 2D cultures were further assessed with the WST-1 reagent (Roche, Basel, Switzerland). The assay was performed in 48-well culture plates with 4 × 10^3^ cells per well. All of the cells were cultured in 100 µL DMEM (high glucose and GlutaMAX + 20% FBS + 10 mM Hepes Buffer + 4x Non-Essential Amino Acids). The cells were incubated at 37 °C and 5% CO_2_ for 48 h for U87-MG and HT7606 and 96 h for SVGp12 to grow to 80–90% confluence. To perform the assay, 10 µL of the WST-1 reagent was added to each well. After an incubation time of 2 h, the plate was read at 440 nm using a microplate reader. 

### 4.12. Three-Dimensional Growth Assay

We performed a spheroid formation and volume growth assay using the 96-well plate liquid overlay technique as described earlier [55]. In brief, the 3-D aggregation capacity of U87-MG and U87-MG-IDH1^R132H^ was compared by seeding the defined cell concentrations into non-adherent 96-well plates coated with 1.5% agarose in DMEM. The plates were incubated according to 2-D culture conditions and the spheroid formation, integrity, circularity, and size were routinely monitored by phase contrast imaging after a four-day initiation interval and every 48–72 h thereafter for up to 20 to 50 days. The spheroids were fed in parallel by a 50% medium exchange. A total of ≥20 individual spheroids were recorded as a function of time. The 3-D volume growth kinetics were mathematically modelled via the Gompertz function as highlighted in [55,56].

### 4.13. Clonogenic Survival

Clonogenic survival assays were carried out as described earlier to assess the colony formation capacity and radio-response [57]. In brief, single cells (250–4000 cells/well) were seeded into 6-well plates and incubated for 4 h to allow sufficient cell adherence before single dose irradiation with 0–12 Gy (200 kV X-rays; 0.5 mm Cu filter; YxlonY.TU 320, Yxlon International, Hamburg, Germany). The cells were then cultured under standard conditions for 9–14 days (depending on the cell type to guarantee ≥6 doublings), and colonies with ≥50 cells were manually counted upon fixation and staining with Coomassie-blue. The surviving fractions (SF) were normalized for cell-line and treatment-dependent differences in colony formation and plating efficiency, respectively, to quantify radio-response. The data were reproduced (N = 3 with n ≥4 wells per experiment and treatment condition), and cell survival curves were fitted employing the linear-quadratic model SF = exp − (αD + βD^2^) with D being the dose and α,β as variables defining the irradiation dose.

### 4.14. Statistical Analysis

The statistical analysis was performed using GraphPad Prism 6 (GraphPad Software, Inc., San Diego, CA, USA). All data are expressed as mean ± SD if not stated otherwise. Statistical analyses were conducted using either one-way analysis of variance (ANOVA) followed by Dunnett’s post-hoc *t*-test, when more than two conditions were compared, or a *t*-test, when only two conditions were compared. Linear regression was applied for the analysis of cell survival curves and for the comparison of the variables (α, β) of the linear-quadratic model using SPSS Statistics 21 (IBM Corporation, NY, USA). The statistical significance was inferred at *p* < 0.05.

### 4.15. TCGA Data Analysis

The gene expression analysis of the key enzymes involved in NAD-synthesis was done using publicly available glioma samples from The Cancer Genome Atlas (TCGA) database. We downloaded processed RNA-seq normalized read counts from the TCGA data portal (gdc.cancer.gov) for 54 astrocytomas WHO grade II, 83 astrocytomas WHO grade III, and 348 primary glioblastomas WHO grade IV. All of the considered astrocytoma samples were reported to have an IDH1/2 mutation. The vast majority of the glioblastoma samples were from patients with primary glioblastomas that lack IDH1/2 mutations. We compared the gene-specific expression levels of glioblastomas against those in astrocytomas using Wilcoxon rank sum tests. 

## 5. Conclusions

In summary, we demonstrated the 2-HG independent effects of IDH1^R132H^ on metabolism, redox state, and energy homeostasis and showed that these effects differ between glial cells and tumor cells. Moreover, we found that cells depend on NAD^+^ to compensate for the IDH1^R132H^-mediated drop in NADPH and that IDH1^R132H^ influences the expression of the NAD^+^ synthesizing enzyme NAMPT. The impaired redox state and energy homeostasis evoked by the IDH mutation has the potential to serve as a therapeutic target in the future.

## Figures and Tables

**Figure 1 cancers-11-02028-f001:**
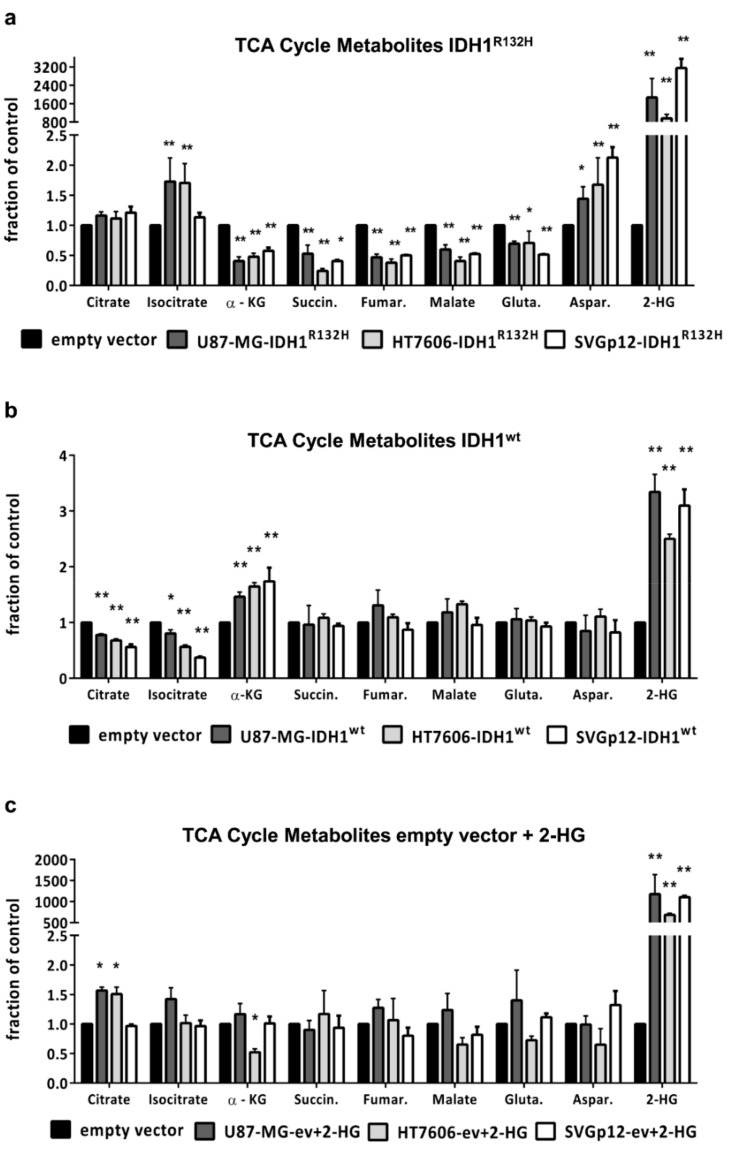
IDH1^R132H^ influences intracellular TCA (tricarboxylic acid) cycle metabolite levels: Concentrations of TCA metabolites were measured using liquid chromatography-tandem mass spectrometry (LC-MS/MS) and results are shown relatively to the empty vector control for (**a**) cells transduced with IDH1^R132H^, (**b**) cells transduced with IDH1^wt^, or (**c**) empty vector controls cells treated with 1 mM D-2-Hydroxyglutarate. For better overview, Y-values were converted to fractions of control. The control of each cell line was defined as a 1.0 baseline and the Y-values were divided by the baseline. All statistical analyses were performed on untransformed data comparing IDH1^R132H^, IDH1^wt^, or empty vector+ 2-HG to empty vector cells using one-way analysis of variance (ANOVA) followed by Dunnett’s post-hoc *t*-test. (* *p* < 0.05, ** *p* < 0.01).

**Figure 2 cancers-11-02028-f002:**
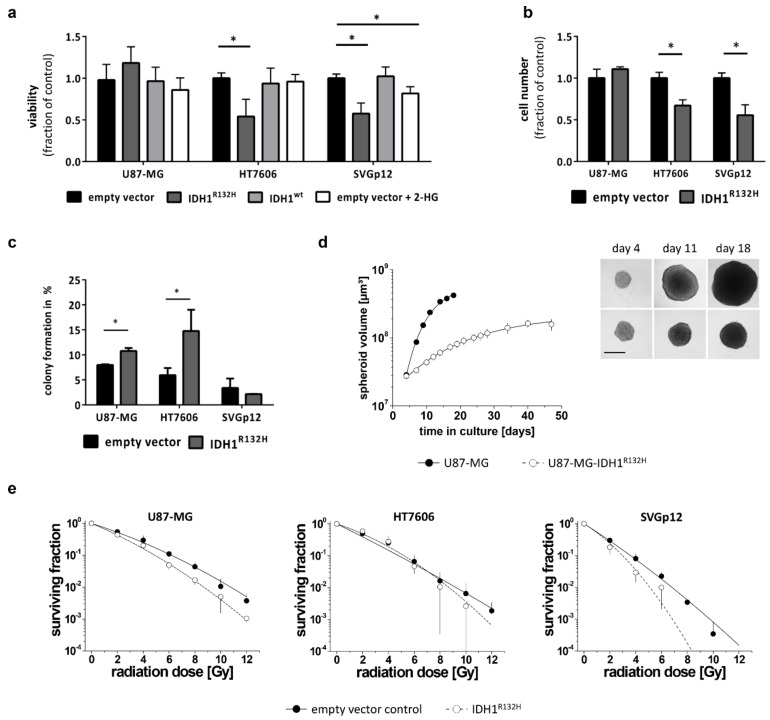
IDH1^R132H^ reduces growth and increases radio-sensitivity: (**a**) Cell viability was determined using a WST-1 based colorimetric assay after 48 h culture in adherent condition. Values were normalized to empty vector cells and means from all experiments performed with different transductions were compared (* *p* < 0.01; one-way analysis of variance (ANOVA) followed by Dunnett’s post-hoc *t*-test). (**b**) Quantification of cell numbers counted using CASY^®^ TTC 72 h after seeding. (**c**) Clonogenic survival assays showed that IDH1^R132H^ significantly enhanced the capacity of glioblastoma cells to form colonies, but not of the astrocytes. (**d**) To analyze cell growth under 3-D conditions, U87-MG control and IDH-mutant cells were seeded in liquid overlay and cultured for up to 50 days. Spheroid size and volume were routinely monitored. In the example shown here, 2 × 10^3^ cells/well were seeded. Data are expressed as the mean volume ± SD. Representative phase contrast microscopic images of the same spheroids are shown at day 4, 11, and 18 (scale bare = 400 µm). (**e**) Radiation-dose response curves were derived from clonogenic survival assays. Data are expressed as mean of three biological experiments ± SD with n ≥4 wells per experiment and treatment condition. Dose response curves were fitted using a linear-quadratic model (surviving fraction = exp − (αD + βD^2^)).

**Figure 3 cancers-11-02028-f003:**
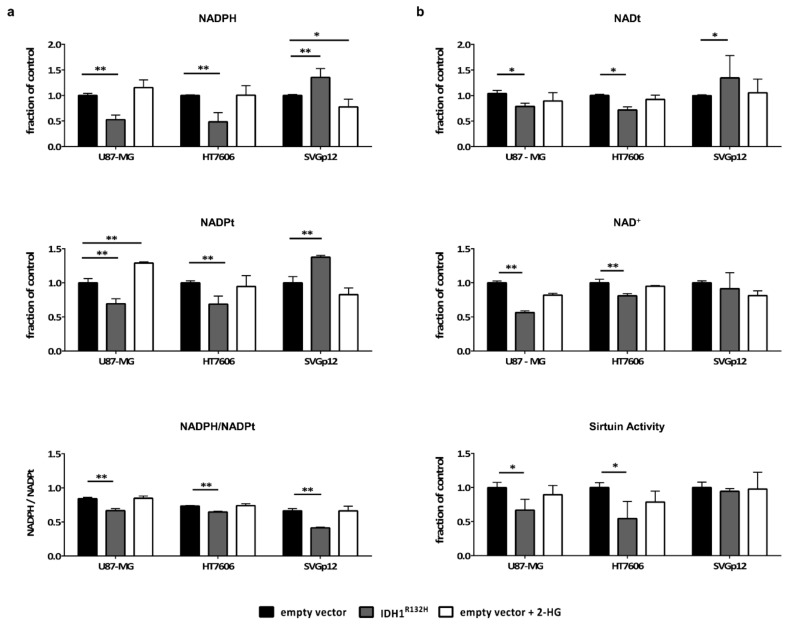
IDH1^R132H^ and not 2-HG alone leads to a drop in NADPH and NAD^+^ concentrations and sirtuin activity in glioblastoma cells but not in astrocytes: Concentrations of NADPH/t and NAD^+^/t were measured in cell lysates of stably transduced cell lines from three different transductions and in triplicates using the NAD^+^/NADH and NADP^+^/NADPH Quantification Kit (MBL). The activity of NAD^+^ dependent sirtuins was measured using the HDAC Fluorimetric Cellular Activity Assay Kit (Enzo Life Science). The values were normalized to the mean value of the empty vector cells and the means of normalized values were compared (* *p* < 0.05; ** *p* < 0.01; one-way analysis of variance (ANOVA) followed by Dunnett’s post-hoc *t*-test). (**a**) Normalized values of NADPH levels (top) and NADPt levels (middle) as well as NADPH/NADPt ratios (bottom) are given. (**b**) IDH1^R132H^ led to a significant drop in NADt (top) and NAD^+^ (middle) levels in glioblastoma cells, but not in astrocytes. Sirtuins showed less enzymatic activity in IDH1^R132H^ glioblastoma cells compared to the empty vector cells (bottom).

**Figure 4 cancers-11-02028-f004:**
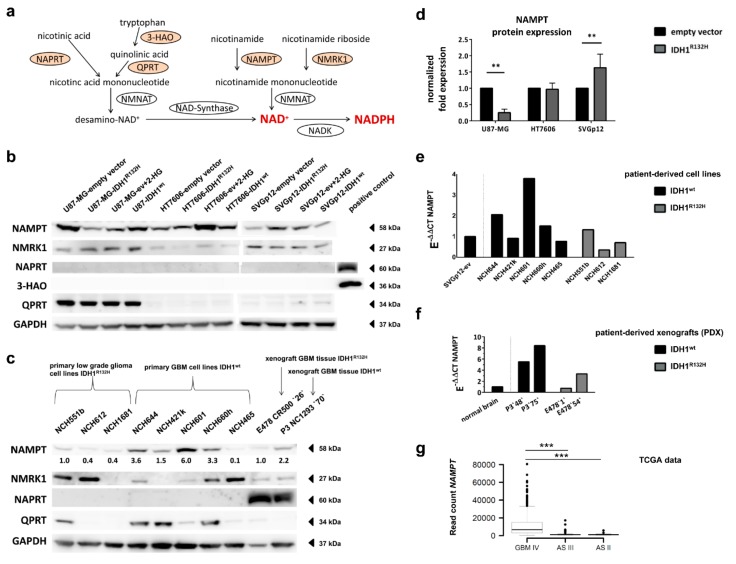
IDH1-mutation influences the expression of the NAD^+^ synthesis enzyme NAMPT: (**a**) NAD synthesis and salvage pathway (adapted from [27]). NAD is synthesized de novo from tryptophan or from nicotinic acid, nicotinamide riboside and nicotinamide via a salvage pathway. NAD-Kinase (NADK) generates NADPH from NAD^+^ and ATP. NAPRT: nicotinicacid phosphorybosiltransferse, NMRK1: nicotinamide riboside kinase, NAMPT: nicotinaminde phosphoribosyltransferase, QPRT: quinolinic acid phosphoribosyltransferase, 3-HAO: quniolinicacid-synthesis-enzyme 3-hydroxyanthranilate 3,4-dioxygenase. (**b**) Representative Western Blot showing the expression of NAD-Synthesis enzymes NAMPT, NMRK1, NAPRT, QRPT and 3-HAO in our cell models. GAPDH was applied to determine protein loading. (**c**) Proteins expression of NAD-Synthesis enzymes in patient-derived cell lines and xenografts (PDX) tissues of IDH1^R132H^-mutant and IDH-wildtype gliomas. To quantify NAMPT protein levels, density per sample was divided by the loading-control (GAPDH) and relative density for that lane is given. (**d**) Mean values of NAMPT protein expression normalized to GAPDH and the empty vector controls from four Western Blots performed with protein extracts of different transductions are shown. (**e**,**f**) NAMPT RNA expression levels were measured using reverse transcription quantitative PCR (RT-qPCR). Gene expression was normalized to the expression of reference genes GAPDH and ARF1 (E-ΔCT) and thereafter to the expression in astrocytes or commercially available RNA from normal brain control tissue (E-ΔΔCT). (**g**) Box plots showing gene expression of NAMPT in IDH1-wildtype (GBM IV) and IDH1-mutant (AS III and AS II) glioma patients based on The Cancer Genome Atlas (TCGA) RNA-seq normalized read count data. Box limits indicate 25th and 75th percentiles, whiskers extend at most to 1.5 times the interquartile range of the box, dots represent outliers. NAMPT was significantly higher expressed in GBM IV versus AS II and III (Wilcoxon rank sum test: *p* = 1.456541 × 10^−26^ and *p* = 1.240315 × 10^−36^, respectively). (** *p* < 0.01, *** *p* < 0.001 one-way analysis of variance (ANOVA) followed by Dunnett’s post-hoc *t*-test).

**Figure 5 cancers-11-02028-f005:**
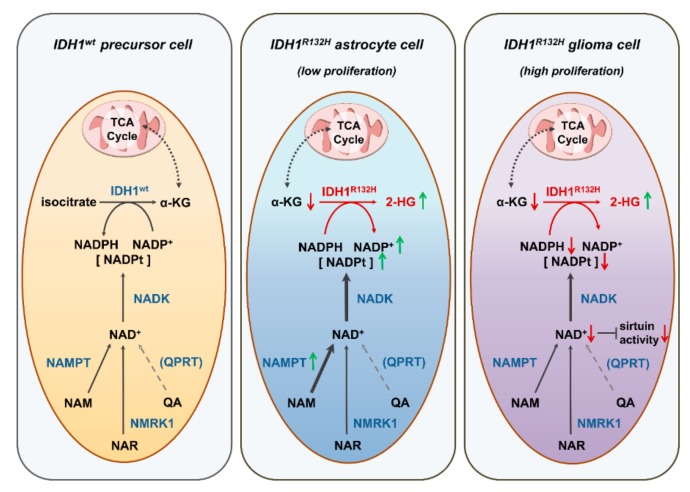
Proposed hypothesis of different effects of the IDH1^R132H^ mutation in glial and tumor cells: We found that the IDH1^R132H^ mutation differently affects the redox state of glial cells and tumor cells. Wildtype IDH1 provides essential amounts of NADPH for the cell whereas IDH1^R132H^ consumes α-KG and NAPDH, leading to abnormally high concentrations of 2-HG, reduced concentrations of α-KG and downstream TCA cycle metabolites, as well as an imbalance between NADPH and NADP^+^ levels. Based on our observations, we hypothesize that in astrocytes, the increased NADPH consumption by IDH1^R132H^ can still be compensated for by elevating the total NADP pool via the induction of NADK and the NAD^+^ synthesizing enzyme NAMPT. Malignant, proliferating cells, however, cannot compensate for the imbalance of NADPH/NADP^+^ due to IDH1^R132H^, leading to decreased NADPH and NAD^+^ levels. This could be due to the increased requirement of NAD^+^ and/or NADPH in proliferating cells or insufficient upregulation of NAD synthesis pathways, potentially accompanied by additional inhibition of NAMPT expression due to the IDH1^R132H^. Abbreviations: NAD = nicotinamide adenine dinucleotide, NADP = nicotinamide adenine dinucleotide phosphate, NMRK1 = nicotinamide riboside kinase, NAMPT = nicotinaminde phosphoribosyltransferase, QPRT = quinolinic acid phosphoribosyltransferase, NADK = NAD-Kinase.

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
