# Peer review of "Mutant IDH1 Differently Affects Redox State and Metabolism in Glial Cells of Normal and Tumor Origin"

_cancers, 2019, doi:10.3390/cancers11122028_

Round 1

Reviewer 1 Report

It was a pleasure to read such an interesting and well-written study. I have no further comments

Author Response

We thank reviewer 1 for the favourable review.

Reviewer 2 Report

IDH mutations play a key role in the development of low-grade gliomas. 2HG production in gliomas with mIDH1/2 mutations have been subject of intense research with many studies demonstrated using cell cultures, animal xenografts, as well as in vivo imaging studies.  

Authors studied redox state and metabolism in the presence of IDH1 mutations. They have demonstrated that IDH1 mutation directly affects energy homeostasis and redox-state in a cell type dependent manner. Targeting altered metabolism and redox state might be an interesting new avenue for treatment of IDH1-mutant gliomas. They also validated using in vivo tissue of patient derived xenografts with and without IDH1 mutations.

The manuscript is well written, illustrated and concise. The data are presented in a clear manner and are appropriately interpreted. This study advances the field of Neuro-oncology significantly.

Minor points: 1) it is helpful to mention about other IDH1 mutations other than mIDH1R132H and discuss why they chose to focus on R132H mutation. 2) Is there any invivo imaging techniques that can be used to study the altered redox state and metabolism? if possible, it will be helpful to discuss this. 

Author Response

We thank reviewer 2 for revising and discussion of our manuscript.

Minor points:

1) it is helpful to mention about other IDH1 mutations other than mIDH1R132H and discuss why they chose to focus on R132H mutation.

The IDH1 R132H mutation is by far the most common in glioma, noted in >90% of cases (Miller et al., Cancer 2017). Therefore, IDH1 R132H is in the focus of our work. As pointed out by reviewer 2, there have been other variants described in IDH1, including R132C, R132S, R132G, and R132L (Hartmann et al, Acta Neuropathol. 2009), which show a similar change in IDH1 activity and result in the accumulation of 2-HG (Dang et al., Nature 2009). As suggested by reviewer 2, we added this information in our introduction, line 59 to 61 and adapted lines 84 and 86.

 2) Is there any in vivo imaging techniques that can be used to study the altered redox state and metabolism? if possible, it will be helpful to discuss this.

Indeed, there are in vivo imaging techniques to study altered metabolism. High levels of 2-HG in IDH1-mutant tumors can be detected using in vivo magnetic resonance spectroscopy (MRS). MRS for 2-HG offers a method with which to monitor IDH‐mutant tumors longitudinally. A decrease in 2-HG levels detected by MRS correlates with clinical and radiologic response (Miller et al., Cancer 2017). It was also possible to show decreased glutamate levels in IDH1 mutant gliomas using MRS (Nagashima et al., Neuro Oncol 2016). We added these points to our discussion, line 335 to 337.

Reviewer 3 Report

The authors characterized metabolic and redox changes in IDH1-mutated glioblastoma and astrocytes cells in vitro. They demonstrated that IDH1 R132H astrocytes were able to increase NAMPT expression to maintain NAD+ and restore NADPH pools, while glioblastomas were not. Using TCGA published transcriptomics patients cohort, they validated that IDH1-mutant gliomas expressed lower NAMPT expression than IDH1-WT gliomas. 

The work is interesting and relevant, and it is complementary of Tateishi et al (Cancer Cell, 2015) study which demonstrated higher sensitivity of IDH1-mutant cells to NAMPT inhibitors in vivo in solid tumors due to NAPRT1 lower expression.

Here, the authors notably investigated the link between NAD+ and NADPH, one of the substrates of the mutated enzyme. They also importantly distinguished the effects of 2-HG and the mutant activity on the redox and metabolic changes observed. Moreover, they evaluated the differences between the impact of the mutation on normal and tumor glial cells.

However, several questions remained unaddressed in the current work.

1/ In Figure 1a, aspartate levels are significantly increased in IDH1 R132H cells in the three models, which could be of particular interest. Could the authors propose hypothesis and address this point in the manuscript?

2/ In Figure 2b and d, could the authors show the data for IDH1 WT and empty vector+2-HG to strenghten the specificity of the findings?

3/ In line 155, the authors stated that "Clonogenic survival assays showed that tumor cell lines were intrinsically more radio-resistant than immortalized astrocytes, with U87-MG exhibiting lowest radio-sensitivity (Figure 2e)". It seems to me that the figure shows the opposite.

4/ Could the authors integrate the radio-sensitivity better with the other findings and conclusions of the study or at least introduce the rationale to measure it here?

5/ It would be interesting to evaluate if IDH1 mutant inhibitor reverses the TCA cycle metabolites amounts, the level of NAD+ and the expression of NAMPT.

6/ Could the authors speculate as to whether similar changes observed with IDH2 are found in gliomas (even if rare)?

Minor Points:

1/ l. 117: 'This indicates that IDH1 R132H negatively affects cell metabolism'; I think the authors should reformulate this sentence as changes in metabolites amounts are not dynamic readouts. They can equally describe an increase or a decrease in the associated fluxes depending on nutrient uptakes, other metabolic pathways, potential inhibitions of specific enzymes... These results show that TCA cycle is impacted but not necessarily 'negatively'.

2/ Figures 2c and 2d should be switched to be consistent with the text.

3/ It would be better if each biological replicate was shown in the graphs of the different figures instead of bars to evaluate the variability. For example, there are discrepancies between Figure 3 (all replicates) and Figure S2 (just one replicate).

Author Response

We thank reviewer 3 for revising our manuscript and critical discussion of our findings.

Our point-by-point response:

1/ In Figure 1a, aspartate levels are significantly increased in IDH1 R132H cells in the three models, which could be of particular interest. Could the authors propose hypothesis and address this point in the manuscript?

We absolutely agree with reviewer 3, that the observed significant increase in aspartate levels are an interesting finding. Indeed, to the best of our knowledge this increase of aspartate has not been described or analysed in the context of IDH1-mutation. High levels of aspartate may provide evidence for the upregulation of an α-KG supplementation pathway in IDH1 R132H mutated cells. The enzyme Glutamic-oxaloacetic transaminase (GOT) uses glutamate and oxaloacetate as substrates to catalyse the reaction to aspartate and α-KG. IDH1 R132H mutated cells could enhance this metabolic reaction to compensate the loss of α-KG caused by the α-KG consuming reaction of IDH1 R132H. Thus, IDH1 R132H mutated cells could overproduce aspartate as a byproduct of the GOT reaction to aim for α-KG supplementation from glutamate. However, this remains a hypothesis, and further investigations are needed to understand this metabolic alteration in IDH1 R132H mutated cells. We included the finding of increased aspartate levels in the text (results line 116-118) and discussed the hypothesis of increased glutaminolysis via GOT as possible explanation for the observed metabolic changes in the discussion section line 336 to 350.

2/ In Figure 2b and d, could the authors show the data for IDH1 WT and empty vector+2-HG to strengthen the specificity of the findings?

To test if the IDH1mut affects viability, we initially performed 2D viability assay using WST-1 based colorimetric assay in our different cell models and compared cells transduced with IDH1mut vs IDH1empty vector (Figure 2a). In this experiments, we also included as additional conditions cells transduced with IDHwt and cells treated with 2-HG, to exclude that overexpression of IDH(wt) or treatment with 2-HG alone would have an effect (Figure 2a). We found that transduction with IDHmut reduced viability in HT7606 and SVGp12 but not in U87 compared to empty vector control, while transduction with IDHwt did not have an effect in all cell lines. To validate that IDHmut affects viability in HT7606 and SVGp12 but not U87 in 2D, we did additional experiments using cell counts after 72h (Figure 2b), which confirmed the finding in Fig 2a. Furthermore, to exclude that the observed reduction in viability/overall growth upon transduction with IDH1mut might be due to reduced colony forming capacity, we additionally performed colony formation assays comparing again cells transduced with IDHmut and empty vector (no Figure 2c – former figure 2d – see minor comment below). For this confirmation experiments, we however did not included the additional conditions, since there were no clear effects in the first experiments and we were here interested in the effect of IDHmut. Moreover, adding 2-HG daily for 50 days for the colony formation would have not been suitable. Therefore, unfortunately, we cannot provide these data.

3/ In line 155, the authors stated that "Clonogenic survival assays showed that tumor cell lines were intrinsically more radio-resistant than immortalized astrocytes, with U87-MG exhibiting lowest radio-sensitivity (Figure 2e)". It seems to me that the figure shows the opposite.

We thank the reviewer for pointing this out and apologize for our mistake. Indeed, in Figure 2e, we accidentally switched the curves for astrocytes and U87. Astrocytes are in our experiments the most radio sensitive, while U87 are the most resistant cells, as we described in the text. We corrected Figure 2e accordingly.

4/ Could the authors integrate the radio-sensitivity better with the other findings and conclusions of the study or at least introduce the rationale to measure it here?

The main rational to measure radio-sensitivity in vitro was because it is known that IDH1 mutant glioma compared to their IDH1 wt counterparts show a better prognosis and response to treatment. Therefore, we wanted to analyse if radio-sensitisation can be also observed in our cell models and connect this with metabolic findings. To make this more clear, we adapted the introduction (line 60-61) and introduced this rationale in the result section (line 137-139).

It has been hypothesized that radiosensitivity in IDH mutant tumors is associated with low NADPH levels since NADPH is an essential substrate in protection against reactive oxygen species (ROS). This hypothesis is supported by findings of others, that silencing IDH1 as the main source of NADPH in brain is improving response to radiotherapy (Wahl DR et al, Cancer Res. 2017). Importantly, we measured a significant drop of NADPH in our tumor cell lines transduced with IDH1 R132H, but not in astrocytes. Nevertheless, astrocytes showed increased radiosensitivity after transduction with IDH1 R132H. Our results therefore indicate that the IDH mutation indeed increases radiosensitivity, but this might not only be explained by lower NADPH levels. Meachanisms behind increased radiosensitivity still need to be elucidated. We addressed the connection of IDH1 mutation, NADPH levels and radio-sensitivity in our discussion line 366 to 380, which we have adapted to better integrate our findings of radio-sensitivity with the literature and our findings regarding NADPH.

5/ It would be interesting to evaluate if IDH1 mutant inhibitor reverses the TCA cycle metabolites amounts, the level of NAD+ and the expression of NAMPT.

We completely agree with reviewer 3 that it would be very interesting to study the effects of IDH1 mutant inhibitors on TCA metabolites and levels of NAD+ and NAMPT in our cell lines and we have planned to include such inhibitors in our future studies. Indeed, the question, if IDH1 mutant inhibitors might even reversing the “negative effects” of the IDH1 mutation itself and therefore may even lead to disease progression and worse therapy response was raised in the literature (Molenaar RJ et al, Cancer Res. 2015) and our models can be useful to address this question in the future.

6/ Could the authors speculate as to whether similar changes observed with IDH2 are found in gliomas (even if rare)?

This is a very interesting point. Compared to IDH1 mutations, mutations in the IDH2 gene occur at an analogous arginine, at position 172 (Yan et al., N Engl J Med. 2009) and, as for IDH1 mutations, the effect on enzyme function seems to be similar. The frequency of this mutation is much less common, accounting for <1% of all IDH mutations in patients with glioma (Hartmann et al, Acta Neuropathol. 2009). As mutant IDH1, mutant IDH2 is unable to efficiently catalyze the oxidative decarboxylation of isocitrate, but acquires a neomorphic catalytic activity that allows the NADPH-dependent reduction of α-KG into 2-hydroxyglutarate (2-HG). It has already been shown that IDH1 and IDH2 mutated glioma produce 2-HG, show a hypermethylated phenotype and increased oxidative stress.  Therefore, one could speculate that IDH2 mutations might lead to similar changes to redox state and metabolism as observed in IDH1 mutated cells.

However, IDH2 is located in the mitochondria and IDH1 is located in cytoplasm. As IDH1 is one of the main NADPH producing enzymes in the cytoplasma besides G6PDH, the loss of NADPH in the cytoplasm can essentially influence cytoplasmic redox status. In contrast, mutated IDH2 would mainly influence mitochondrial redox status. The counter acting mechanisms to preserve redox homeostasis might be different to IDH1 mutated cells, but this is highly speculative. Studies would be necessary that compare IDH1-mutated and IDH2-mutated models with respect to redox-state. To the best of our knowledge, there is currently no data in the literature addressing this question. Therefore, we did not address this point in our discussion. However, it would be indeed very interesting to analyse similarities and differences between IDH1 and IDH2 mutated glioma cells with respect effects on energy metabolism and cytoplasmatic/mitochondrial redox state in the future. Unravelling potential differences between IDH1 and IDH2 mutations might help to understand why IDH2 mutations are much rarer in gliomas, which is currently unclear.

Minor Points:

1/ l. 117: 'This indicates that IDH1 R132H negatively affects cell metabolism'; I think the authors should reformulate this sentence as changes in metabolites amounts are not dynamic readouts. They can equally describe an increase or a decrease in the associated fluxes depending on nutrient uptakes, other metabolic pathways, potential inhibitions of specific enzymes... These results show that TCA cycle is impacted but not necessarily 'negatively'.

We thank reviewer for pointed this out. Indeed the used method does not give dynamic readouts and therefore we removed the word “negatively” in this sentence (line 123).

2/ Figures 2c and 2d should be switched to be consistent with the text.

We switched the order of Figure 2c and 2d, exchanged it with the new figure in the manuscript and adapted the text accordingly. Therefore, former Figure 2c is now 2d in the text and the legend of Figure 2 was adapted.

3/ It would be better if each biological replicate was shown in the graphs of the different figures instead of bars to evaluate the variability. For example, there are discrepancies between Figure 3 (all replicates) and Figure S2 (just one replicate).

Several repetitions using biological and technical replicates for the different cell lines (biological replicates: cell lines were transduced with the same protocol but at different time points; experiments were repeated for biological replicates; within each experiment, technical replicates were used) were performed. Using biological replicates, we intended to reduce intra- and inter-experimental variabilities. The effects of IDH1 R132H mutation on NADPH/NADP/NAD/sirtuin-activity were therefore analysed using the data of all biological replicates and experiments, rather than one experiment by itself. Figure 3 shows this combined result (mean values) for all experiments and the inter-experimental variability is shown as variability bars. Repetition experiments for the different cell models/different biological replicates were not necessarily done at the same time. Showing all individual experiments separately would take many individual figures and the overall results as well as variability between experiments could not be easily observed anymore.
In supplementary Figures S2, we nevertheless show exemplarily one individual experiment, where we analysed all three different cell lines at the same time in one experiment to illustrate the differing absolute levels of NADPH and NADPt between the different cell lines instead of the levels normalized to the control (fraction of control). Small discrepancies in the control 2-HG treatment arm in this particular experiment reflect variability between individual experiments.

Round 2

Reviewer 3 Report

I thank the authors for the responses to all of my comments. I think the revised version provide interesting findings and points of discussion to pursue our understanding of IDH mutations in glioma and other cancers.